# Interactive Buckling of Steel LC-Beams Under Bending

**DOI:** 10.3390/ma12091440

**Published:** 2019-05-03

**Authors:** Zbigniew Kolakowski, Jacek Jankowski

**Affiliations:** Lodz University of Technology, 90-924 Lodz, Poland; jacek.jankowski@p.lodz.pl

**Keywords:** lip channel, bending, interactive buckling, distortional-lateral mode, different lengths

## Abstract

The present paper deals with the interactive buckling of thin-walled lipped channel (LC) beams under the bending moment in the web plane when the shear lag phenomenon and distortional deformations are taken into account. A plate model (2D) was adopted for LC beams. The structures were assumed to be simply supported at the ends. A modal method of solution to the interactive buckling problem within Koiter’s asymptotic theory, using the semi-analytical method (SAM) and the transition matrix method, was applied. LC-beams, from short through medium-long via long to very long beams, were considered. The paper focuses on the influence of the secondary global buckling mode on the load carrying capacity for the steel LC-beams under bending.

## 1. Introduction

Thin-walled cold-formed steel (CFS) members (columns, beams, and beam-columns) are widely used in the construction industry. C-section and LC-section (i.e., lipped channel) beams are basic structural elements that are primarily subject to bending. A capacity for resistant loads in thin-walled beams is limited not just by their strength, but first of all due to their stability.

The numerical methods often applied in nonlinear analysis of stability and load carrying capacity are as follows: finite strip method (FSM), finite element method (FEM), and generalized beam theory (GBT).

Mode decomposition of thin-walled columns and beams with different cross-sections subjected to various loadings based on GBT has been analyzed in Reference [1]. References [2,3] deal with the constrained finite element method (cFEM) employed for the buckling analysis of columns with opened cross-sections. A new method with a modal decomposition feature, the so-called constrained finite element method (cFEM) is presented in Reference [4]. The method can be applied to analyze a wide range of thin-walled members, including members with holes and varying cross-sections and stiffened plates.

FSM is commonly used for nonlinear analysis of elastic stability due to its very high numerical capability, low computational costs and easy implementation in thin-walled elements. The method is limited mainly to simple geometries and boundary conditions. It has resulted in an introduction of numerous new FSM variants or expansions.

In FEM, two kinds of analyses are usually conducted, namely for (i) linear elastic buckling to enable determination of critical loads and the corresponding buckling modes, i.e., an eigenproblem, (ii) nonlinear post-buckling analysis to determine the performance curve in the full range of loading and/or load carrying capacity. In addition, the FEM model is affected by epistemic uncertainties.

The influence of distortional, global, and local buckling modes, and their combination, on post-buckling behavior is widely investigated by Martins, Camotim, Gonçalves, and Dinis [5,6]. They and used their modification (GBTUL-2.0) [5] and employed existing tools (GBT and DSM) [6]. A semi-analytic method based on Koiter’s theory of the compressed thin-walled structure with cross-section deformation modes is analyzed in Reference [7]. Composite C-beams subjected to the bending moment have been analyzed in References [8], among others. The development in the theory of interactive buckling of thin-walled structures is discussed in Reference [9]. Observation of the post-buckling behavior of beams during experimental investigations of channel-section beams subjected to pure bending [10] inspired the research presented here. Moreover, another interpretation of the obtained results has been adopted. Reference [10] that dealt with post-buckling behavior of short beams with channel-section, subjected to pure bending, were the basis to carry out the research presented in this paper. In References [11,12], the nonlinear problem of stability has been solved using the semi-analytical method (SAM) based on Koiter’s nonlinear theory with Byskov-Hutchinson interactive buckling approach. An interaction between two or three buckling modes (i.e., two- and three-mode approach) has been analyzed, whereas in Reference [13] numerous buckling modes were considered. The effect of various lengths (from the short ones through medium-long to long beams) of C-beams subjected to the bending moment, with linearly variable stresses in the web plane, on the interactive buckling and load carrying capacity using semi analytical method (SAM) [11,12] was investigated. The present paper is a continuation of Reference [13], focusing on steel bending LC-beams to confirm the generality of the results obtained in Reference [13] and to specify the length of the beam for which the influence of the secondary mode of buckling on load carrying capacity appears in the range of the most dangerous case. A literature review and the theoretical foundation are presented in detail in Reference [13].

The post-buckling behavior of beams during experimental investigations of channel-section beams subjected to pure bending was described in Reference [14]. Moreover, the influence of the distortional-lateral buckling mode on the interactive buckling of thin-walled short channels with imperfections subjected to the major-axis bending moment was analyzed. With the GBT it is possible to define contributions of different buckling modes in the full range of structure loading, which facilitates understanding the phenomena occurring during complex coupled buckling.

Moreover, composite C-beams subjected to the bending moment have been analyzed in Reference [15], among others. 

A distortional failure of steel beams simply supported under uniform bending with the direct strength method (DSM) is presented in Reference [16]. The influence of distortional, global, and local buckling modes, and their combination, on post-buckling behavior is widely investigated by Martins, Camotim, Gonçalves, and Dinis [17,18]. In Reference [19], the interactive buckling of lipped channel (LC) beams freely supported at both ends and subject to uniform major-axis bending was analyzed with the generalized beam theory (GBT). Special attention was paid to the interaction of very numerous simple buckling modes, and, in particular, to the distortional-global mode effect. Moreover, the influence of distortional, global, and local buckling modes, and their combination, on post-buckling behavior is widely investigated in References [20,21].

In this study, an analysis of LC beams with the same dimensions as in Reference [19] has been conducted. In wide experimental investigations and the numerical analysis of steel C-beams [22,23], particular attention was drawn to the distortional-global interaction buckling. Post-buckling behavior of cold-formed channels axially compressed or subjected to pure bending was analyzed in Reference [24]. In Reference [25], the buckling behavior and imperfection sensitivity of thin steel cylindrical shells under pure bending, with a focus range of slenderness, were presented.

The fire-resistant design of steel columns used in prefabricated modular construction was described in References [26,27].

Modal analysis of interactive buckling enables an easier insight into the phenomena occurring during the interactions of global, local, and distortional buckling modes. Moreover, it allows an interpretation of complex interactions between individual buckling modes.

In the authors’ opinion, there is a lack of work devoted to the influence of the secondary global distortional-lateral mode on the interactive buckling and assessment of the load carrying capacity for LC-beams of various lengths. The paper attempts to assess for what length of beam this impact is the most important. In the authors’ opinion, the literature includes adequate information about the influence of the secondary global distortional-lateral mode on the interactive buckling and assessment of the load carrying capacity of LC-beams of various lengths. The aim of this investigation is to determine what values of length of the beam influences on the load carrying capacity.

In the present study, Lagrange’s description, full Green’s strain tensor for thin-walled plates and second Piola–Kirchhoff’s stress tensor, and the exact transition matrix method and the numerical method of the transition matrix using Godunov’s orthogonalization are used. The shear lag phenomenon, an effect of cross-sectional distortions, as well as coupled conditions between all the walls of structures are included. The most important advantage of this method is that a complete range of behavior of thin-walled structures can be described [14].

The coupled buckling of thin-walled steel LC-beams under bending in the web plane from short ones through medium-long to long beams is analyzed here.

## 2. Formulation of the Problem

Prismatic thin-walled steel (i.e., isotropic) beams built of plates connected along longitudinal edges and under the uniform major-axis bending moment were considered. The beams were simply supported at their ends [13,14]. In order to account for all modes of global, local, and coupled buckling, a plate model (i.e., 2D) of thin-walled structures was applied. Moreover, it was assumed that the material of the structure is obeyed Hooke’s law. Details can be found in References [13,14] or see Appendix A.

## 3. Analysis of the Calculations Results

Detailed numerical calculations for interactive buckling were conducted for three steel LC-section beams of the cross-section dimensions identical to those in Reference [19]. The beam geometrical dimensions under consideration together with the assumed notations are presented in Figure 1 and Table 1. The ratios of the main central moments of inertia *I_max_/I_min_*, which range from 3.6 to 6.0, are included there as well.

The following material constants: *E* = 210 GPa, *ν* = 0.3 were assumed for the steel lip channels [19]. In the pre-buckling state, the beams were subjected to linearly variable loading (Figure 1) resulting in bending in the web plane (i.e., the upper flange was tensioned, the lower one was compressed).

### 3.1. Example of LC-1 Beams

In this example, lip channel LC-1 beams of the dimensions listed in Table 1 were analyzed. Alternations in values of the critical moment *M_r_* as a function of the buckling half-wavelength *L_b_* in a wide variability range 100 ≤ *L_b_* ≤ 10,000 mm are presented in Figure 2.

The lower curve (denoted as curve 1) corresponds to the lowest values of buckling loads, often referred to as primary buckling loads. The upper curve (marked as curve 2) refers to higher critical values, which can be called secondary buckling loads.

The value of the critical moment *M_r_* (curve 1) for the variability range under consideration attains its maximum at *L_b_* = 100 mm and the minimum at *L_b_* ≈ 350 mm, and then it grows monotonously up to *L_b_* ≈ 1000 mm, where it reaches the local maximum. Within the range 1000 ≤ *L_b_* ≤ 10,000 mm, values of the moment decrease monotonously. The critical values corresponding to curve 2 grow in the range 100 ≤ *L_b_* ≤ 350 mm and attain the maximal value at *L_b_* ≈ 350 mm. Next, in the range 350 ≤ *L_b_* ≤ 1000 mm, they decrease drastically to attain the minimal value for *L_b_* ≈ 1000 mm. Within the range 1000 ≤ *L_b_* ≤ 4000 mm, curve 2 grows slowly, and at *L_b_* > 4000 mm, it is constant in practice. While comparing curves 1 and 2, one can state that for *L_b_* ≈ 350 mm the lower curve attains its minimum, whereas the upper one attains its maximum, and then at *L_b_* ≈ 1000 mm, an opposite relation takes place. At *L_b_* > 4000 mm, the drop gradient of curve 1 is significantly lower than for 1000 ≤ *L_b_* ≤ 4000 mm, whereas the values corresponding to curve 2 are actually constant.

In Table 2, critical values of the moments *M_r_* for the LC-1 subject to bending for selected four values of the total length *L* are presented. The following index notations are introduced: 1—the lowest value of the buckling moment corresponding to the local buckling mode for *m* ≠ 1, 2—the value of the primary global buckling mode for *m* = 1 (curve 1 in Figure 2), 3—the value of the secondary global buckling mode for *m* = 1 (curve 2 in Figure 2). For the local moment *M*_1_, a number of half-waves *m* along the longitudinal direction is quoted additionally in the brackets.

For the lengths of LC-1 under consideration, the values of the critical moment *M*_1_ do not alter significantly (less than 10%). The values of *M*_3_ are at least sixfold higher than *M*_1_. At the length *L* ≈ 2050 mm, the value *M*_1_ ≈ *M*_2_, whereas, at *L* = 1500 mm, we have *M*_2_/*M*_1_ = 1.6, and for *L* = 500 mm it is *M*_2_/*M*_1_ = 1.16. At the length *L* = 250 mm, the lowest critical value was attained for the local mode *M*_1_ and for one buckling half-wave (*m* = 1). Therefore, the critical value *M*_2_ corresponding to the global mode was not given. The value *M*_3_ is almost 14 times higher than *M*_1_ and also occurs for *m* = 1.

In Figure 3a–d, for the lengths of LC-1 beams considered in Table 2, the buckling modes corresponding to the three modes under analysis, except *L* = 250 mm, for which only two modes (i.e., mode 1 and mode 3) are considered, are shown.

For the three shortest lengths, the local mode 1 (*r* = 1) is the same (Figure 3a–c). The upper corner connecting the flange under tension with the web and the edge reinforcement does not displace practically. The maximal deflection corresponds to the compressed flange corner with the edge reinforcement. At the length *L* = 2050 mm (Figure 3d), the local mode is different. The maximal deflection occurs for the lower part of the web under compression, and both corners of the compressed flange displace. Local modes correspond to distortional-local buckling modes.

At *L* = 500 mm (Figure 3b), the lowest global mode (at *r* = 2) is identical to the local mode. For the lengths *L* = 1500 mm and *L* = 2050 mm (Figure 3c,d), the global mode represents distortional-lateral buckling because right angles are not maintained in the corners of cross-sections of the elements under compression. The secondary global mode (*r* = 3) is subject to alternations with an increase in the length of LC-1. For *L* = 200 mm, the maximal deflection occurs for the web. Only the corner connecting the lower edge reinforcement with the flange under compression displaces (Figure 3a). For other lengths, the lower corner connecting the flange with the web also displaces and there is a slight displacement of the corner connecting the web with the flange under compression. At *L* = 500 mm (Figure 3b), the maximal displacement corresponds to lower corners, whereas, at *L* = 1500 mm and *L* = 2050 mm (Figure 3c,d), the modes differ slightly. The secondary global modes correspond more to the distortional-global modes than the distortional-lateral ones.

In the nonlinear analysis of interactive buckling, the signs of complex absolute values of imperfections of each mode were selected in the safest way, i.e., to attain the lowest value of the limit load carrying capacity *M_s_* [11,12,13,14] in (A4). For actual LC-section beams, post-buckling equilibrium paths were determined on the assumption in (A4) that $\zeta_{1}^{*}=\left|0.1\right|$, $\zeta_{2}^{*}=\left|1.0\right|$, $\zeta_{3}^{*}=\left|1.0\right|$.

For the lengths *L* under consideration, Table 2 also lists values of the limit load carrying capacity referring to the lowest value of the critical moment *M_min_* = *M*_1_, and accounts for *M*_*s*1_/*M_min_* for a three-mode approach (i.e., *J* = 3 in (A4)) and *M*_*s*2_/*M_min_* for a two-mode approach (i.e., *J* = 2). At *L* = 250 mm, due to the fact that both modes occur for *m* = 1, it was assumed on the contrary that $\zeta_{3}^{*}=\left|0.1\right|$. For this length, interaction between buckling modes (denoted by indices *r* = 1 and *r* = 3) does not take place within the loading range *M/M_min_* under analysis.

In Figure 4, on the basis of Equation (A6), a plot of *M/M_min_* versus the angle $\alpha /\alpha_{\min}$ is presented. Curve 1 corresponds to a one-mode analysis, that is to say, when only the mode *J* = *r* = 1 is considered, whereas curve 2 corresponds to a two-mode analysis for *J* = 2 (for *r* = 1 and *r* = 3).

These curves overlap in the range of variability of *M/M_min_* under analysis. At *L* = 500 mm (Table 2 and Figure 5), when the interaction of the three modes is taken into account, we have the limit value of *M*_*s*1_/*M_min_*, whereas, for an interaction of two modes (i.e., *J* = 2 for *r* = 1 and *r* = 2), the theoretical limit load carrying capacity was not obtained.

As shown in Figure 5, curve 1 corresponds to the interaction of three modes, whereas curve 2 corresponds to the interaction of two modes, respectively. For *L* = 1500 mm, *M*_*s*1_/*M_min_* is approximately 2% lower than for *M*_*s*2_/*M_min_*. The lowest values of load carrying capacity were attained at *L* = 2050 mm and they are practically the same for the two- and three-mode approach.

The strongest interaction of the local mode (*r* = 1) with the global one (*r* = 2) occurs for the case when the critical loads are close to each other, i.e., when the relationship 0.8 ≤ *M*_2_*M*_1_ ≤ 1.2 holds. When *M*_2_/*M*_1_ ≈ 1, as known from the literature, the load carrying capacity often satisfies the relation 0.6 ≤ *M_s_M*_1_ ≤ 0.7. For *L* = 2050 mm, *M*_2_/*M*_1_ = 1.007 and *M*_*s*1_/*M*_1_ ≈ 0.675 occur and, for *L* = 1500 mm, *M*_2_/*M*_1_ = 1.60 and *M*_*s*1_/*M*_1_ = 0.773 occur, correspondingly, whereas, for *L* = 500 mm, *M*_2_/*M*_1_ = 1.16 and *M*_*s*1_/*M*_1_ ≈ 0.768 occur, respectively. As can be expected, an interaction of three modes yields lower values of the load carrying capacity than an interaction of two modes.

An interaction of buckling modes [11,12] takes place via the coefficients of cubic terms ${\bar{a}}_{pqr}\zeta_{p}\zeta_{q}\zeta_{r}$ in the expression for total potential energy (A3). Thus, values of the coefficients ${\bar{a}}_{pqr}$ for all lengths *L* under study were analyzed. For a short beam of *L* = 250 mm, the terms including the coefficients $\zeta_{1}^{2}\zeta_{3}$ in (A3) are very low and buckling can be treated as uncoupled (i.e., one-mode) for the loads *M/M_min_* under consideration. It is also due to very considerable differences in values of critical loads, because *M*_3_/*M*_1_ ≈ 14. At *L* = 500 mm, the terms $\zeta_{1}^{2}\zeta_{3}$, $\zeta_{2}^{2}\zeta_{3}$ decide the interaction, whereas, for *L* = 1500 mm and *L* = 2050 mm, these are the terms $\zeta_{1}^{2}\zeta_{2}$, $\zeta_{1}^{2}\zeta_{3}$.

In Reference [19], local imperfections were taken as in the present work, whereas global imperfections were assumed for selected buckling modes, and their level was close to that assumed here. In Reference [19], for various global imperfections and at *L* = 2050 mm, *M_s_/M_min_* = 0.864 was attained, and, in this work, *M*_*s*1_/*M_min_* = 0.675 was attained.

### 3.2. Example of LC-2 Beams

The geometrical dimensions of the LC-2 beam are listed in Table 2. Figure 6 shows a change in the critical bending moment *M_r_* [MNcm] as a function of the buckling half-wavelength *L_b_* in the range 100 ≤ *L_b_* ≤ 10,000 mm.

Curve 1 corresponds to the lowest critical values of the bending moment, i.e., the primary buckling moments, whereas curve 2 corresponds to the secondary buckling moments. Curve 1 decreases in the range 100 ≤ *L_b_* ≤ 400 mm, and then it increases up to the maximal value at *L_b_* = 1500 mm. For higher lengths *L_b_*, the critical moment decreases monotonously. On the other hand, curve 2 grows monotonously for 100 ≤ *L_b_* ≤ 500 mm to attain its maximal value at *L_b_* = 500 mm. At 500 ≤ *L_b_* ≤ 1500 mm, it decreases sharply to grow next in the range 1500 ≤ *L_b_* ≤ 4000 mm, and then the critical values remain constant for *L_b_* ≥ 4000 mm in point of fact.

To sum up, curve 1 attains its local minimum at *L_b_* ≈ 400 mm, curve 2 has its maximum at *L_b_* ≈ 500 mm, curve 1 attains its maximum and curve 2 its minimum at *L_b_* ≈ 1500 mm.

In Table 3 the results for the assumed four total lengths of LC-2 beams are collected. The index notations were the same as in example 3.1 (LC-1). For the assumed lengths, values *M*_1_ are lower and are actually the same except for the shortest beam of the length *L* = 250 mm. At *L* = 250 mm and *L* = 400 mm, the moment *M*_1_ corresponds to the number of half-waves *m* = 1. Thus, as for LC-1, the values of *M*_2_ corresponding to the global mode are not given. On the other hand, the values of *M*_3_ for the secondary mode, at which *m* = 1, are given. For the assumed lengths *L*, we have *M*_3_/*M*_1_ > 6, and, for two longest beams, it is *M*_2_/*M*_1_ ≥ 1.5. Hence, for the lengths under consideration, sensitivity to imperfections decreases in comparison to example 3.1 (LC-1).

In Figure 7a–d, buckling modes for LC-2 are shown. The local buckling mode (mode 1) for the four assumed lengths is the same.

The deflection maximum lies in the corner of the compressed flange and reinforcement. The secondary global mode (mode 3) for *L* = 250 mm and *L* = 400 mm is the same. The maximal deflection takes place in the web. At *L* = 2000 mm, mode 3 is similar to the local mode. Additionally, only displacements of the corner connecting the web with the flange under compression can be observed. At *L* = 700 mm and mode 3, the maximal deflection of the web is slightly higher than the displacement of lower corners. For this length, the global mode (curve 2) is identical to the local mode (curve 1), while for *L* = 2000 mm, mode 2 corresponds to the distortional-lateral buckling mode, as there are no right angles in lower corners. Thus, all buckling modes (curves 1, 2, 3) are distortional modes.

In Table 3, the values of the ratio of the limit load carrying capacity to the minimal critical value for two- (*J* = 2) and three- (*J* = 3) mode approaches, *M*_*s*2_/*M_min_* and *M*_s1_/*M_min_*, respectively, are given. Like in example 3.1, the same values of imperfections were assumed.

For the lengths *L* = 250 mm and *L* = 400 mm, limit values were not attained. For these lengths as for LC-1, it was assumed that $\zeta_{3}^{*}=\left|0.1\right|$, as *m* = 1. For the remaining two lengths, the limit load carrying capacity is lower for the three-mode approach than for the two-mode approach, identically as for LC-1.

In the two next figures (Figure 8 and Figure 9), a relationship of *M/M_min_* versus the angle *α/α_min_* is presented according to formula (A6) for the length *L* = 250 mm and *L* = 400 mm.

Curve 1 corresponds to the case of one-mode buckling (*r* = *J* = 1), while curve 2 corresponds to two-mode buckling (*J* = 2, *r* = 1, *r* = 3). Both the curves overlap, which proves a lack of an interaction between the modes in the range of loading under consideration. The dependence of *α/α_min_* on *M/M_min_* at *L* = 700 mm, for the two- (*J* = 2) and three-mode (*J* = 3) approach, correspondingly, is presented in Figure 10. In the case of *J* = 3, the limit load carrying capacity is *M*_*s*1_/*M_min_* = 0.867, whereas, for *J* = 2, *M*_*s*2_/*M_min_* cannot be determined.

In this case, a significant effect of the secondary global mode (*r* = 3) on the load carrying capacity can be seen. At *L* = 2000 mm, the quantities *M*_*s*1_/*M_min_* and *M*_*s*2_/*M_min_* differ slightly, i.e., by less than 2%.

For *L* = 250 mm and *L* = 400 mm, the nonlinear coefficients (A3) $\zeta_{1}^{2}\zeta_{3}$, responsible for the interaction of modes, are very low and, moreover, *M*_3_/*M*_1_ > 6; thus, we encounter one-mode buckling for the loads *M/M_min_* under analysis. At *L* = 700 mm, the terms including the coefficients $\zeta_{1}^{2}\zeta_{3}$, $\zeta_{2}^{2}\zeta_{3}$, $\zeta_{3}^{2}\zeta_{2}$ play an important role, whereas, at *L* = 2000 mm, the terms are $\zeta_{1}^{2}\zeta_{2}$, $\zeta_{1}^{2}\zeta_{3}$.

In Reference [19], for the length *L* = 2000 mm, the dimensional load carrying capacity is equal to *M_s_/M_min_* = 0.919, and, in this work, it is *M*_*s*1_/*M_min_* = 0.803. One should remember that the values of global imperfections were assumed differently. At *L* = 400 mm in Reference [19], the load carrying capacity was not determined either.

### 3.3. Example of LC-3 Beams

Like in earlier examples, detailed geometrical dimensions are listed in Table 2. In Figure 11, alternations in critical bending moments *M_r_* as a function of the buckling half-wavelength *L_b_* are presented.

Curve 1 corresponds to the lowest values of the critical moment, while curve 2 corresponds to higher values for 100 ≤ *L_b_* ≤ 10,000 mm. The plots of both curves are similar to the plots in Figure 2 (LC-1) and Figure 6 (LC-2). The minimal local value of the moment for curve 1 was attained at *L_b_* ≈ 450 mm, the local maximum was attained at *L_b_* ≈ 1500 mm, whereas curve 2 attains the maximal value of the moment for *L_b_* ≈ 480 mm and the minimal value for *L_b_* ≈ 1500 mm, respectively.

As in former examples, Table 4 lists values of critical loads for 4 selected lengths of beams *L*. At *L* = 300 mm, the lowest local mode *M*_1_ occurs for *m* = 1. Thus, mode 2 was not considered. The value *M*_3_ (for *m* = 1) is almost 10-times higher than *M*_1_. At *L* = 800 mm, we have *M*_2_/*M*_1_ = 1.4, and for *L* = 2500 mm it is *M*_2_/*M*_1_ = 1.3, whereas, at *L* = 4500 mm, the global value *M*_2_ is lower than *M*_1_, as *M*_2_/*M*_1_ = 0.5. The secondary value of *M*_3_ for the values of *L* under analysis is at least 7-times higher than *M*_1_.

In Figure 12a–d, buckling modes for selected lengths *L* are presented. Local buckling modes (mode 1) are practically the same for all lengths.

At *L* = 300 mm and mode 3 (*m* = 1), maximal deflections occur in the web. At *L* = 800 mm, also the global mode (mode 2) is identical to mode 1 (Figure 12b). The secondary global mode (mode 3) has the maximal deflection for lower compressed corners of LC-3. Mode 2 (curve 2) for the length *L* = 2500 mm and *L* = 4500 mm is a “pure” lateral buckling mode in principle. At *L* = 2500 mm and mode 3, a slight displacement of the corner connecting the web with the compressed lower flange takes place, whereas, for *L* = 4500 mm, displacements of both web corners occur.

Moreover, Table 4 also shows the dimensionless limit load carrying capacity for two- and three-mode approaches, *M*_*s*2_/*M_min_* and *M*_*s*1_/*M_min_*, respectively. At *L* = 2500 mm and *L* = 4500 mm, differences between both the approaches are inconsiderable.

In Figure 13, at the length *L* = 300 mm, curve *1* for the one-mode approach (*J* = 1) overlaps curve 2 for the two-mode approach (*J* = 2, *r* = 1, *r* = 3).

For the range of loadings under consideration, there is no interaction between the modes. For *L* = 800 mm, in the case of the three-mode, we have *M*_*s*1_/*M_min_* = 0.833, whereas, for the two-mode approach, there is no limit load carrying capacity (Figure 14).

For *L* = 300 mm, the value of the coefficient at the term $\zeta_{1}^{2}\zeta_{3}$ is inconsiderable, but at *L* = 800 mm, the terms $\zeta_{1}^{2}\zeta_{3}$, $\zeta_{2}^{2}\zeta_{3}$ play a significant role. At *L* = 2500 mm and at *L* = 4500 mm, the coefficients at the terms $\zeta_{1}^{2}\zeta_{2}$, $\zeta_{1}^{2}\zeta_{3}$ are important.

For the length *L* = 4500 mm in Reference [19], the value of the load carrying capacity was *M_s_/M_min_* = 0.806, whereas, in the present analysis, it was *M*_*s*1_/*M_min_* = 0.77.

For all the examples under analysis in Reference [19], higher values were attained than in the present study. One should note once more that the values of local imperfections in Reference [19] were assumed in a different way than here.

The plots of variability in critical moments (curves 1 and 2) as a function of the half-wavelength *L_b_* shown in Figure 2, Figure 6 and Figure 11 as well as an analysis of buckling modes, the load carrying capacity and the effect of nonlinear coefficients at the first-order approximation terms allow one to classify, according to the conclusions expressed in Reference [13], the following lengths of LC-beams, namely:
(1)short beams (LC-1 − 100 ≤ *L_b_* ≤ 350 mm; LC-2 − 100 ≤ *L_b_* ≤ 500 mm; LC-3 − 100 ≤ *L_b_* ≤ 450 mm);(2)medium-long beams (LC-1 − 350 ≤ *L_b_* ≤ 1050 mm; LC-2 − 500 ≤ *L_b_* ≤ 1500 mm; LC-3 − 450 ≤ *L_b_* ≤ 1050 mm);(3)long beams (LC-1 − 1050 ≤ *L_b_* ≤ 3500 mm; LC-2 − 1500 ≤ *L_b_* ≤ 4000 mm; LC-3 − 1050 ≤ *L_b_* ≤ 4000 mm);(4)very long beams (LC-1 − 3500 mm ≤ *L_b_*; LC-2 − 4000 mm ≤ *L_b_*; LC-3 − 4000 mm ≤ *L_b_*).

Compared to Reference [13], the term of very long beams, for which the secondary global mode *M*_3_ is actually constant and the primary global mode *M*_2_ has a low gradient of the value drop in comparison to long beams, is introduced additionally in the above-mentioned classification.

Particular attention was paid to the influence of secondary global distortional-lateral buckling mode on the load carrying capacity for the LC-beams under bending. As demonstrated in the paper, the most significant influence is for medium-long beams. In this case, disregarding the interaction of three modes, including two global (i.e., primary and secondary) and local ones, may lead to an incorrect assessment of the load carrying capacity of the two-mode approach for medium-long beams.

## 4. Conclusions

The stability and interactive buckling of steel LC-beams under bending, for three different cross-sections in a wide range of beam length variability, were investigated. Attention was paid in particular to an effect of the secondary global buckling mode on an interaction between modes, including distortional buckling modes. A classification of lengths of beams subjected to bending in the web plane, starting from short to medium-long up to long, or even very long ones, is proposed. In the cases under consideration, the influence of the secondary global mode on the load carrying capacity is most evident for medium-long beams. It is advisable to extend this analysis onto an effect of the length of edge reinforcements and the wall thickness of LC-beams on the interactive buckling and load carrying capacity. It is important to conduct validation of the described and analyzed phenomenon, using the semi-analytical method SAM through the use of more comprehensive modeling using FEM.

## Figures and Tables

**Figure 1 materials-12-01440-f001:**
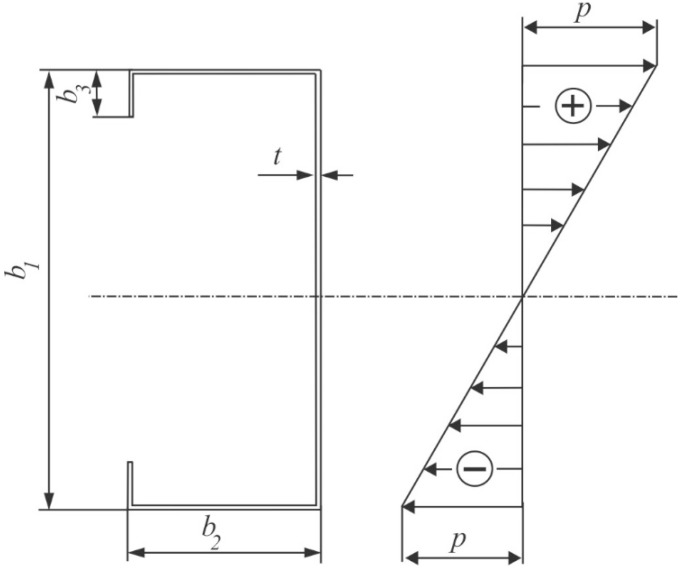
Cross-section of the lip channel.

**Figure 2 materials-12-01440-f002:**
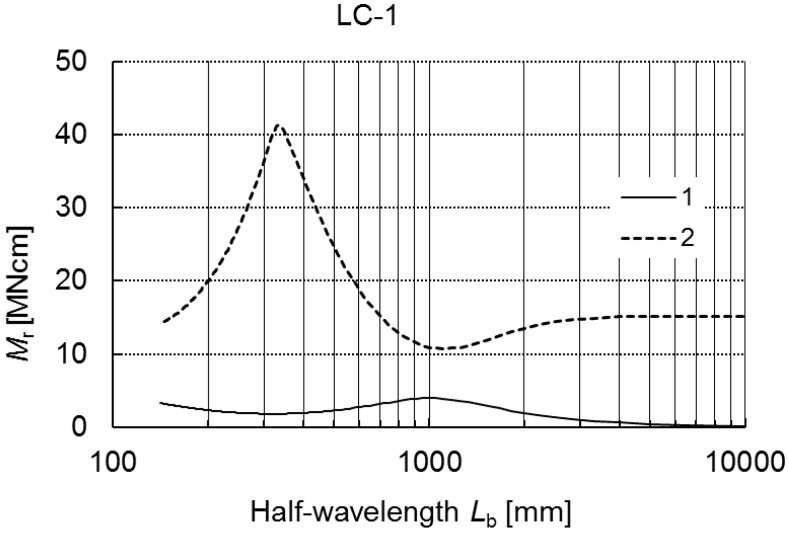
Buckling moments *M_r_* as a function of the buckling half-wavelength *L_b_* for LC-1.

**Figure 3 materials-12-01440-f003:**
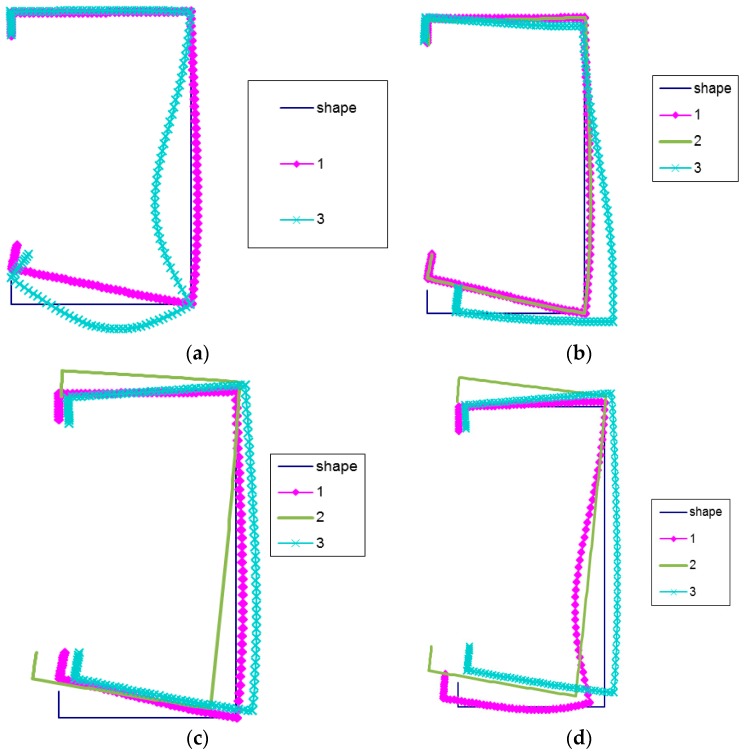
(**a**) Buckling modes for LC-1 beam lengths *L* = 200 mm, (**b**) Buckling modes for LC-1 beam lengths *L* = 500 mm, (**c**) Buckling modes for LC-1 beam lengths *L* = 1500 mm, (**d**) Buckling modes for LC-1 beam lengths *L* = 2050 mm.

**Figure 4 materials-12-01440-f004:**
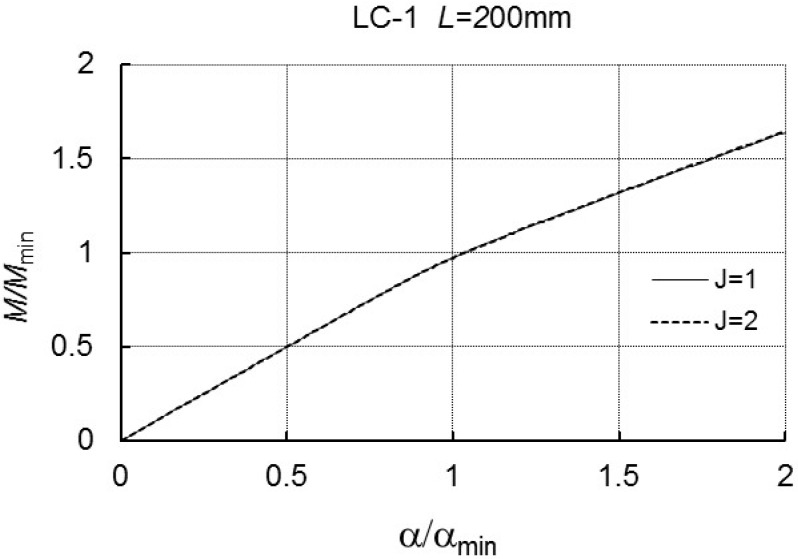
*M/M_min_* as a function of *α/α_min_* for LC-1 and *L* = 200 mm.

**Figure 5 materials-12-01440-f005:**
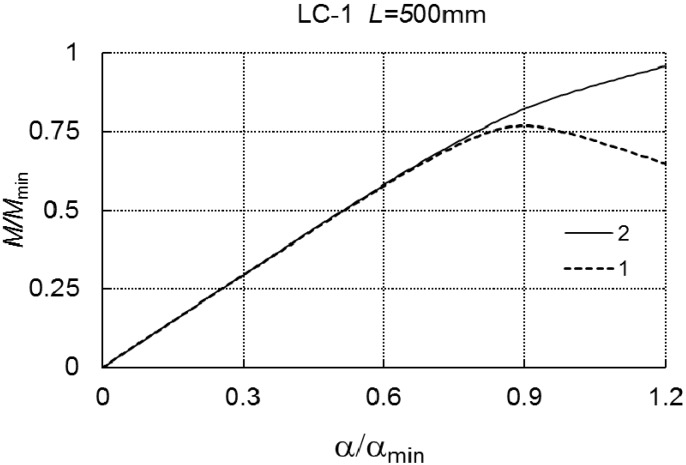
*M/M_min_* as a function of *α/α_min_* for LC-1 and *L* = 500 mm.

**Figure 6 materials-12-01440-f006:**
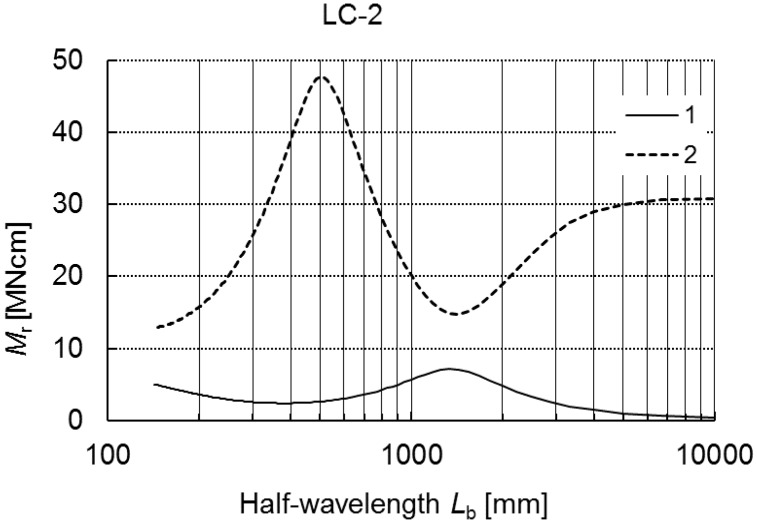
Buckling moments *M_r_* as a function of the buckling half-wavelength *L_b_* for LC-2.

**Figure 7 materials-12-01440-f007:**
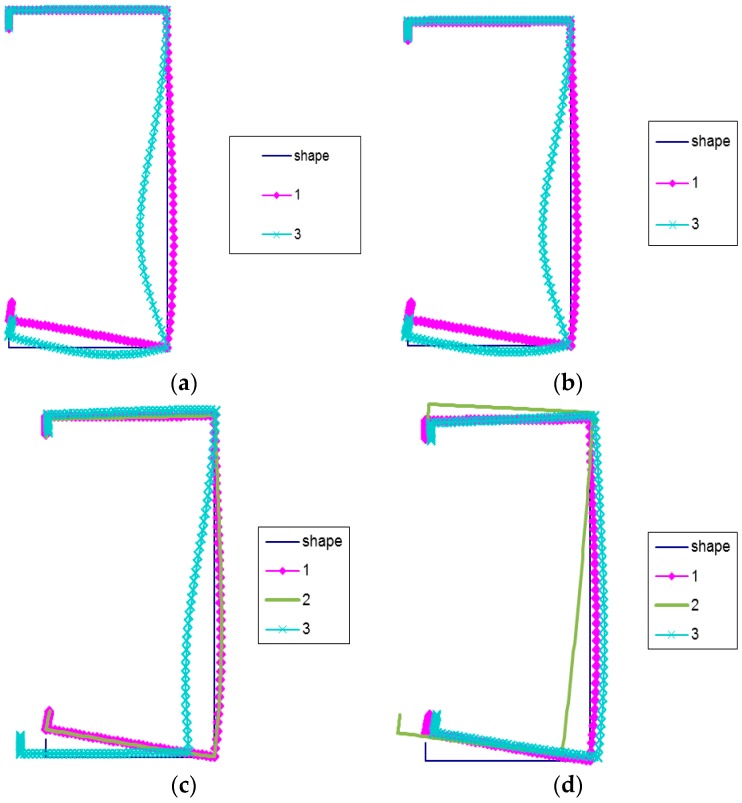
(**a**) Buckling modes for LC-2 beam lengths *L* = 250 mm, (**b**) Buckling modes for LC-2 beam lengths *L* = 400 mm, (**c**) Buckling modes for LC-2 beam lengths *L* = 700 mm, (**d**) Buckling modes for LC-2 beam lengths *L* = 2000 mm.

**Figure 8 materials-12-01440-f008:**
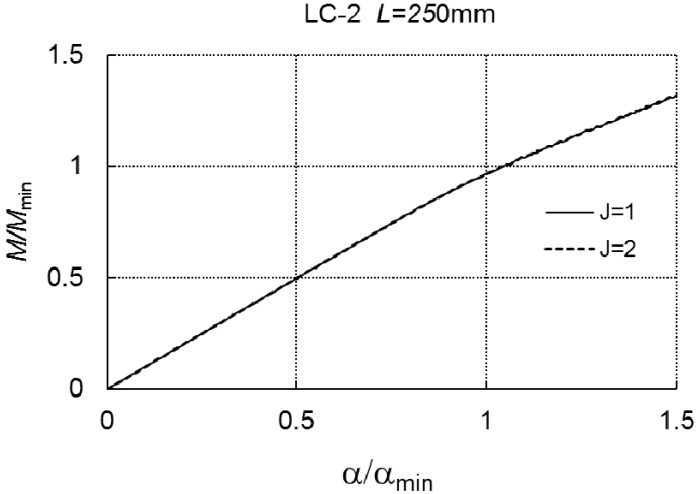
*M/M_min_* versus *α/α_min_* for LC-2 and *L* = 250 mm.

**Figure 9 materials-12-01440-f009:**
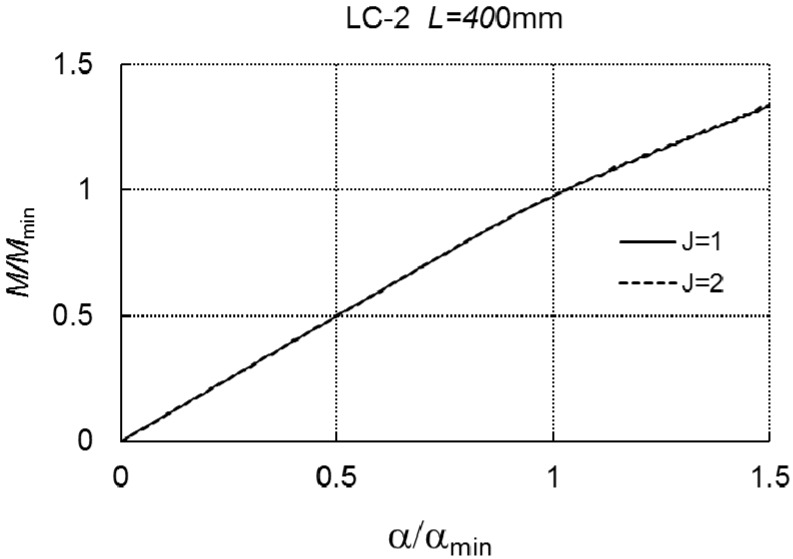
*M/M_min_* versus *α/α_min_* for LC-2 and *L* = 400 mm.

**Figure 10 materials-12-01440-f010:**
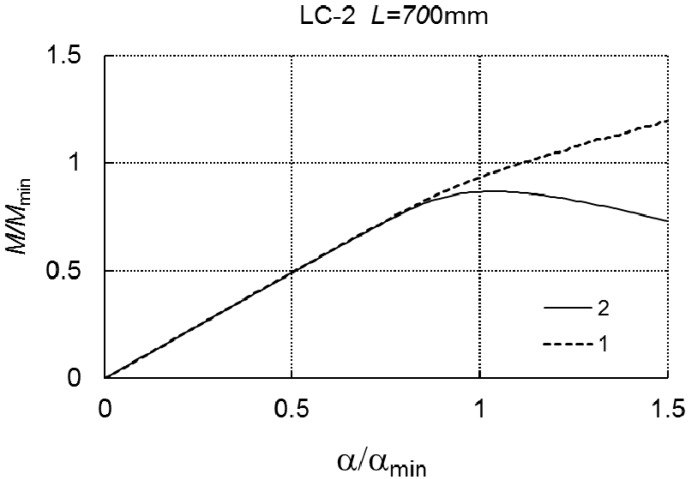
*M/M_min_* versus *α/α_min_* for LC-2 and *L* = 700 mm.

**Figure 11 materials-12-01440-f011:**
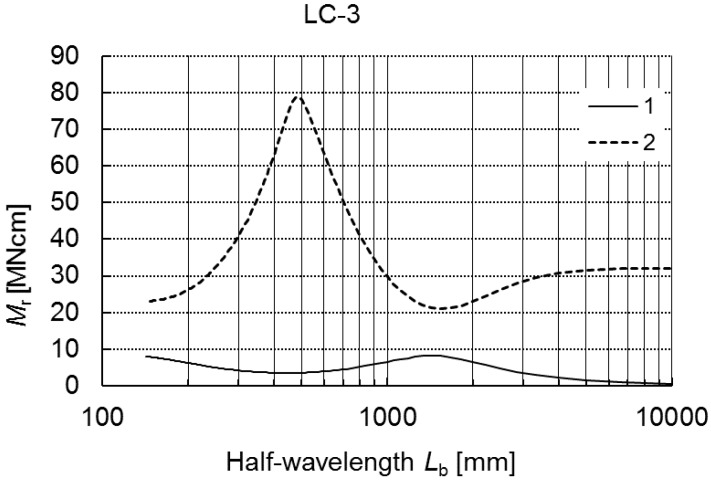
Buckling moments *M_r_* as a function of the buckling half-wavelength *L_b_* for LC-3.

**Figure 12 materials-12-01440-f012:**
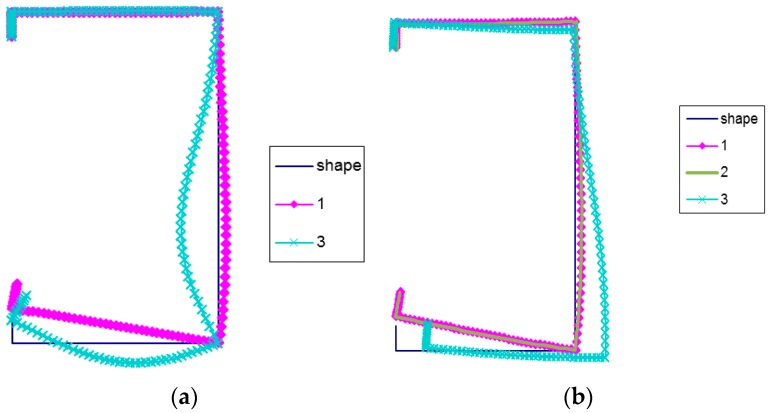
(**a**) Buckling modes for LC-3 beam lengths *L* = 300 mm, (**b**). Buckling modes for LC-3 beam lengths *L* = 800 mm, (**c**) Buckling modes for LC-3 beam lengths *L* = 2500 mm, (**d**) Buckling modes for LC-3 beam lengths *L* = 4500 mm.

**Figure 13 materials-12-01440-f013:**
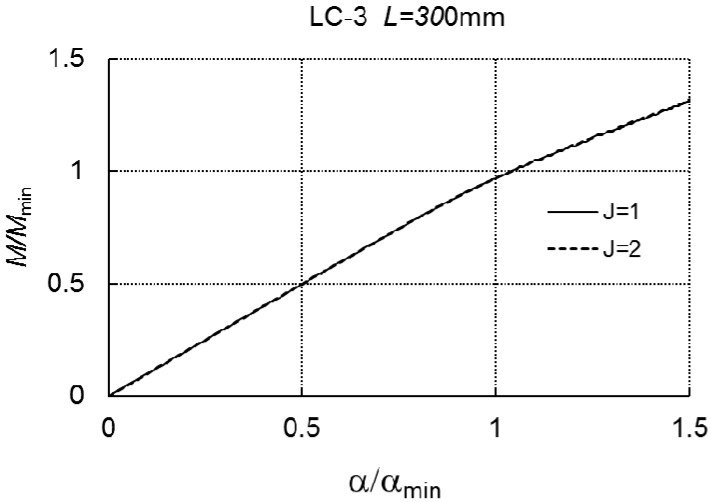
*M/M_min_* versus *α/α_min_* for LC-3 and *L* = 300 mm.

**Figure 14 materials-12-01440-f014:**
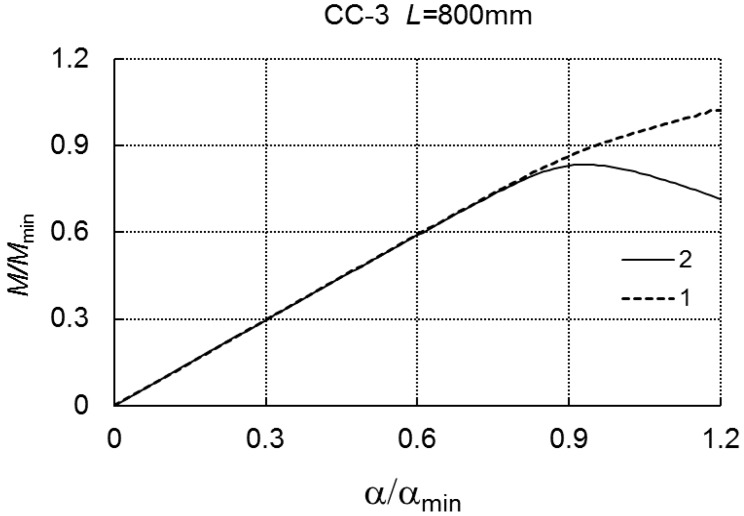
*M/M_min_* versus *α/α_min_* for LC-3 and *L* = 800 mm.

**Table 1 materials-12-01440-t001:** Geometrical dimensions considered lipped channel (LC)-beams.

Type of the Beam	*b* _1_	*b* _2_	*b* _3_	*t*	*I_max_/I_min_*
	[mm]	[mm]	[mm]	[mm]	-
LC-1	125	75	10	3	3.67
LC-2	190	90	10	3.08	5.94
LC-3	175	100	13	3.6	4.04

**Table 2 materials-12-01440-t002:** The buckling moments *M_r_* with the corresponding number of half-waves *m* of buckling along the longitudinal direction of the LC-1 and the dimensionless ratio of load carrying capacities *M_s_/M_min_* for different lengths *L*.

*L*	*M* _1_	*M* _2_	*M* _3_	*M*_*s*1_/*M_min_*	*M*_*s*2_/*M_min_*
mm	MNcm	MNcm	MNcm	-	-
2050	1.575 (6)	1.587	11.64	0.675	0.680
1500	1.574 (5)	2.525	10.13	0.773	0.791
500	1.683 (2)	1.962	21.07	0.768	-
250	1.683 (1)	-	23.42	-	-

**Table 3 materials-12-01440-t003:** The buckling moments *M_r_* with the corresponding number of half-waves *m* of buckling along the longitudinal direction of the LC-2 and the dimensionless ratio of load carrying capacities *M_s_/M_min_* for different lengths L.

*L*	*M* _1_	*M* _2_	*M* _3_	*M*_*s*1_/*M_min_*	*M*_*s*2_/*M_min_*
Mm	MNcm	MNcm	MNcm	-	-
2000	2.413 (5)	4.807	18.94	0.803	0.816
700	2.423 (2)	3.625	34.48	0.867	1.077
400	2.413 (1)	-	38.91	-	-
250	2.913 (1)	-	20.13	-	-

**Table 4 materials-12-01440-t004:** The buckling moments *M_r_* with the corresponding number of half-waves *m* of buckling along the longitudinal direction of the LC-3 and the dimensionless ratio of load carrying capacities *M_s_/M_min_* for different lengths *L*.

*L*	*M* _1_	*M* _2_	*M* _3_	*M*_*s*1_/*M_min_*	*M*_*s*2_/*M_min_*
mm	MNcm	MNcm	MNcm	-	-
4500	3.499 (10)	1.765	31.23	0.770	0.774
2500	3.515 (6)	4.686	26.27	0.753	0.763
800	3.544 (2)	5.105	41.15	0.833	-
300	4.174 (1)	-	40.87	-	-

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
