# Peer review of "Interactive Buckling of Steel LC-Beams Under Bending"

_materials, 2019, doi:10.3390/ma12091440_

Round 1
Reviewer 1 Report
This article has been greatly improved based on the review comments and can therefore be accepted for publication.
Author Response
First of all we would like to thank the Reviewer for his valuable remarks, which all let us improve our papers. Additionally, we have done our best to improve our article. We hope that newest version of our paper fulfills the requirements of publishable standard.
Reviewer 1
Reviewer’s comments No. 1.
This article has been greatly improved based on the review comments and can therefore be accepted for publication.
Authors’ answer:
The text was once again carefully checked and corrected.
Conclusions were filled in, by addition:
‘It is important to conduct validation of described and analysed phenomenon, using semi-analytical method SAM through use of more comprehensive modelling using FEM.’
With best regards,
Zbigniew Kolakowski
Jacek Jankowski

Reviewer 2 Report
Comments to the Authors
I have reviewed the manuscript “Interactive buckling of steel LC-beams under bending” by Zbigniew Kolakowski and Jacek Jankowski. The work is well-organised and needs some revisions to be published in Materials. Here are my comments.
1. Revise some typos in the text.
2. Explain how the initial imperfections are implemented.
3. Within FEM model Section, comment that a FEM model is affected by epistemic uncertainties.
Author Response
First of all we would like to thank the Reviewer for his valuable remarks, which all let us improve our papers. Additionally, we have done our best to improve our article. We hope that newest version of our paper fulfills the requirements of publishable standard.
Reviewer 2
Reviewer’s comments No. 1.
Revise some typos in the text.
Authors’ answer:
The text was once again carefully checked and corrected.
Reviewer’s comments No. 2.
Explain how the initial imperfections are implemented.
Authors’ answer:
It was done in Appendix.
Reviewer’s comments No. 3.
Within FEM model Section, comment that a FEM model is affected by epistemic uncertainties.
Authors’ answer:
It was done in Introduction.
With best regards,
Zbigniew Kolakowski
Jacek Jankowski

Reviewer 3 Report
The paper is an extension of authors' prevoious work (reference entry 13) about interactive buckling of beams.
Despite Authors' effort, the present article does not present any added value but just some further numerical investigations. This is also confirmed by the fact that the whole theoretical part is presented in an appendix. Validation is completely missing. Finally, the article is not properly written (English must be improved).
In this form the paper cannot be supported for publications.
Authors could try to:
1- extend their method by considering, for istance, a functually graded material or buckling due to an assumed thermal field so to provide the article with some scientific interest.
2- add some verifications using a reference solution available in literature or by a FEM package.
3- put the new theoretical part after the introduction
Author Response
First of all we would like to thank the Reviewer for his valuable remarks, which all let us improve our papers. Additionally, we have done our best to improve our article. We hope that newest version of our paper fulfills the requirements of publishable standard.
Reviewer 3
Reviewer’s comments No. 1.
Despite Authors' effort, the present article does not present any added value but just some further numerical investigations. This is also confirmed by the fact that the whole theoretical part is presented in an appendix. Validation is completely missing. Finally, the article is not properly written (English must be improved).
Authors’ answer:
Presented article is the second one that is introduction to observed, during experimental research, rapid exhaustion of load carrying capacity of LC beam under bending. Conducted further numerical validation had in view assurance about regularity of conclusion in the first article. Moreover, numerical validation indicated length of LC beam, for that influence of the secondary global mode on load carrying capacity is the most significant. English correction of article was made also.
Reviewer’s comments No. 2.
Authors could try to extend their method by considering, for instance, a functually graded material or buckling due to an assumed thermal field so to provide the article with some scientific interest.
Authors’ answer:
The article includes analysis only new phenomena, observed for isotropic material, so that, authors does not take under consideration FGM materials and thermal loadings additionally.
Reviewer’s comments No. 3.
Authors could try to add some verifications using a reference solution available in literature or by a FEM package.
Authors’ answer:
In conclusions, a statement was included:
‘There is important to conduct validation of described and analysed phenomenon using semi-analitycal method SAM by use of more comprehensive modelling FEM’.
Reviewer’s comments No. 4.
Authors could try to put the new theoretical part after the introduction.
Authors’ answer:
In authors’ opinion, readability of the article does not require the new theoretical part.
With best regards,
Zbigniew Kolakowski
Jacek Jankowski

Round 2
Reviewer 2 Report
The manuscript is suitable for publication in its current form.
Reviewer 3 Report
Article accepted.
This manuscript is a resubmission of an earlier submission. The following is a list of the peer review reports and author responses from that submission.
Round 1
Reviewer 1 Report
I´d strongly recommend validating the implemented formulation against FEM or similar well-established theory, and presenting the comparison of results (numerically and graphically) before addressing the analysis and conclusions presented in the submitted manuscript.
That said, its not up to me to decide whether or not the paper should be published in current form. Keep up the great job!
Author Response
Reply to the Reviewer's comments
‘Interactive buckling of steel LC-beams under bending’
by Zbigniew Kolakowski, Jacek Jankowski
First of all we would like to thank the Reviewer for his valuable remarks, which all let us improve our papers. Additionally, we have done our best to improve our article. We hope that newest version of our paper fulfills the requirements of publishable standard
With best regards,
Zbigniew Kolakowski
Jacek Jankowski

Reviewer 2 Report
This paper investigated the stability and interactive buckling of steel LC-beams under bending for three different cross-sections in a wide range of beam length variability. This paper has been well written and the conclusions obtained in this work may be useful for engineering practice. Here are the comments:
1. As the authors described, the C-section and LC-section beams are common structural elements, so this reviewer suggests that the research significance and innovation of this article need to be elaborated in the text.
2. There seems to be some typos on page 1, lines 6 & 15, and the format of abstract is not correct.
3. The information of corresponding author should be added.
4. The part of literature review is insufficient with only 15 references, and this reviewer recommends that some recent literatures about thermal performance of building components can also be added to enrich the research background: Materials 2018, 11, 437; Steel Compos. Struct. 2013, 15, 163–173.
5. The langue of this article needs to be further polished, and some figures (e.g. figs. 8-11, 13-14) seem to need a higher resolution.
6. The conclusion should be illustrated in the form of bullet points. This reviewer suggests that some definite values should be added to the conclusion so that engineers can use straight away.
Author Response

(The authors gave the same response as above.)

Reviewer 3 Report
The present work deals with the interactive buckling of thin-walled lipped channel beams under the bending moment. The current version of the manuscript is not sufficient for publication.
1. The abstract of this paper is very similar to that of Ref. (9,10), this problem also exists in the theoretical part of the article. Please elaborate on the similarities and differences between this paper and Ref (9,10).
2. Reference numbering should start from [1].
3. The shear lag phenomenon and distortional deformations are taken into account in this paper, how about this two are not taken into account?
4.The overall scientific merits of this work is limited, as the theoretical background has been published in the previous papers by the same authors. This paper deals with a problem with minor difference of boundary condition.
Author Response

(The authors gave the same response as above.)

Round 2
Reviewer 3 Report
I am still not clear about the difference between this work and Ref [9-11]. If the model is the same, then there is no need to introduce the model again. If the authors want to discuss the influence of certain deformation mode, the organization then needs change.
Author Response
First of all we would like to thank the Reviewer for his valuable remarks, which all let us improve our papers. Additionally, we have done our best to improve our article. We hope that newest version of our paper fulfills the requirements of publishable standard.
The paper is the introduction to describe influence of secondary global distortional-lateral mode on the interactive buckling and the load carrying capacity for various length of LC-beams. Presented paper is the second one of series of articles dedicated analysis of rapid descent of load carrying capacity phenomenon that was observed during experimental research [6,10]. Authors do not know other articles in literature of the subject that are dedicated this phenomenon. Therefore this is introduction to wide analysis that phenomenon.
Below we provide answers to the remark.
Reviewer 3
Reviewer’s comments No. 1.
I am still not clear about the difference between this work and Ref [9-11]. If the model is the same, then there is no need to introduce the model again. If the authors want to discuss the influence of certain deformation mode, the organization then needs change.
Authors’ answer:
Chapter 2 has been significantly shortened. In order to improve the readability of the paper, the Appendix has been added.
With best regards,
Zbigniew Kolakowski
Jacek Jankowski